# From Hospital to Community: Exploring Antibiotic Resistance and Genes Associated with Virulence Factor Diversity of Coagulase-Positive *Staphylococci*

**DOI:** 10.3390/antibiotics12071147

**Published:** 2023-07-04

**Authors:** Hazem Aqel, Naif Sannan, Ramy Foudah

**Affiliations:** 1Basic Medical Sciences Department, College of Medicine, Al-Balqa Applied University, Salt 19117, Jordan; 2King Abdullah International Medical Research Centre, King Abdulaziz Medical City, Jeddah 22384, Saudi Arabia; sannann@ksau-hs.edu.sa; 3Clinical Laboratory Sciences Department, College of Applied Medical Sciences, King Saud Bin Abdulaziz University for Health Sciences, Jeddah 21423, Saudi Arabia; 4Clinical Laboratory Sciences Department, College of Applied Medical Sciences, King Saud Bin Abdulaziz University for Health Sciences, Riyadh 14611, Saudi Arabia; foudahr@ksau-hs.edu.sa

**Keywords:** *Staphylococcus aureus* prevalence, non-hospital personnel carriers, hospital personnel carriers, methicillin-resistant *Staphylococcus aureus*, antibiotic susceptibility testing, genes associated with virulence factors

## Abstract

Coagulase-positive staphylococcus (CoPS), including methicillin-resistant *Staphylococcus aureus* (MRSA), poses a global threat. The increasing prevalence of MRSA in Saudi Arabia emphasizes the need for effective management. This study explores the prevalence of virulence-associated genes and antibiotic resistance patterns in CoPS. Nasal swabs from 200 individuals were collected, and standard protocols were used for the isolation, identification, and characterization of CoPS and coagulase-negative staphylococci (CoNS). Additionally, antimicrobial susceptibility testing and PCR were conducted. Bacterial growth was observed in 58.5% of participants, with 12% positive for CoPS and 30% positive for CoNS. Hospital personnel carriers showed a significantly higher proportion of CoNS compared with non-hospital personnel carriers. Non-hospital personnel CoPS strains displayed higher sensitivity to oxacillin than hospital personnel strains. Cefoxitin exhibited the highest sensitivity among β-lactam antibiotics. All isolates were sensitive to trimethoprim/sulfamethoxazole, rifampin, and quinupristin. Polymerase chain reaction analysis detected methicillin resistance genes in both non-hospital and hospital personnel MRSA strains. The *coa* and *spa* genes were prevalent in MRSA isolates, while the *Luk-PV* gene was not detected. A high prevalence of CoPS and CoNS was observed in both non-hospital and hospital personnel carriers. Occupational risk factors may contribute to the differences in the strain distribution. Varying antibiotic susceptibility patterns indicate the effectiveness of oxacillin and cefoxitin. Urgent management strategies are needed due to methicillin resistance. Further research is necessary to explore additional virulence-associated genes and develop comprehensive approaches for CoPS infection prevention and treatment in Saudi Arabia.

## 1. Introduction

Coagulase-positive *Staphylococcus* (CoPS), particularly *S. aureus*, is a major cause of hospital- and community-acquired infections worldwide [1,2]. These infections can range from minor skin infections to severe cases such as bacteremia, endocarditis, and necrotizing pneumonia that lead to high morbidity and mortality rates [3,4]. The emergence of antibiotic-resistant strains, particularly methicillin-resistant *S. aureus* (MRSA), has posed significant challenges to the treatment of these infections [5,6].

The identification of MRSA occurred in 1961, with a subsequent epidemic observed in 1980. MRSA is a significant global bacterial pathogen in various countries including Saudi Arabia [5,7,8]. The incidence of non-hospital personnel (NHP)- and hospital personnel (HP)-acquired MRSA carriers has increased since 2000 [9,10]. MRSA is a normal flora of the upper respiratory tract, particularly the nose and nasopharynx, and frequently leads to both NHP and HP carriers [10]. Methicillin resistance is attributed to the expression of the *mecA* gene, which alters penicillin-binding protein (PBP-2) to PBP-2a, resulting in a loss of target affinity [11,12]. In addition to antibiotic resistance, CoPS can produce various virulent factors, including virulence-associated genes that contribute to the pathogenesis of infection. These virulence-associated genes include, but are not limited to, Panton–Valentine leukocidin (*PVL*), staphylococcal enterotoxins, and toxic shock syndrome toxin (TSST-1) [13,14,15]. The prevalence of genes associated with virulence factors in *S. aureus*, such as *PVL*, *spa*, *coa*, *aae, aap*, *emb*, and *IcaD*, can have important implications for infection control strategies and treatment approaches. Although these genes themselves may not directly modify the response to treatment, their presence in MRSA strains has been associated with increased pathogenicity and potential complications in patients [16,17]. First, the presence of these genes associated with virulence factors has been linked to more severe infections and increased virulence potential of MRSA strains. As such, understanding their prevalence helps in identifying strains that may pose a higher risk to patients, allowing for targeted infection control measures to prevent their spread within healthcare settings. This knowledge can aid decision makers in implementing appropriate measures, such as isolation protocols and enhanced hygiene practices, to minimize the transmission of these highly virulent strains [18]. In addition, some of these genes have been implicated in antibiotic resistance mechanisms. For example, the presence of the *mecA* gene, which confers resistance to methicillin and related antibiotics, is an important marker for MRSA. The detection of these antibiotic resistance genes helps in identifying strains that may exhibit resistance to certain antibiotics, which is crucial for guiding appropriate treatment decisions [19]. Knowledge of the prevalence of these genes can aid in selecting effective antimicrobial therapies and avoiding the use of ineffective antibiotics, thus optimizing treatment outcomes [20,21]. Understanding the prevalence of genes associated with virulence factors in MRSA strains is valuable for infection control strategies as it helps in identifying highly virulent strains and implementing targeted preventive measures. Additionally, knowledge of the presence of antibiotic resistance genes guides treatment decisions by ensuring the appropriate selection of effective antibiotics. These factors collectively contribute to providing optimum treatment and reducing the impact of MRSA infections [22].

The first epidemiological report on MRSA in Saudi Arabia was in 1994, from the western region (Jeddah) [23,24]. MRSA accounted for approximately 7.5% of all CoPS isolates studied over three years, with wound sites being the most common source [25]. In subsequent studies, MRSA was found in both NHP and HP carriers, with higher rates of multiple resistance observed among hospital carriers [10,26,27]. The prevalence of MRSA in tertiary care hospitals has been increasing in the kingdom, emphasizing the need for effective management measures [28,29].

This study explored the prevalence of genes associated with virulence factors and antibiotic resistance patterns in CoPS isolated from NHP and HP settings. Understanding gene prevalence in healthcare settings is vital for assessing transmission risks, implementing interventions, and improving treatment outcomes. It helps identify individuals who may benefit from personalized approaches and alternative therapies. In addition, gene prevalence informs the selection of appropriate antimicrobial therapies to address antibiotic resistance concerns, aiding decision makers in tailoring treatment regimens effectively.

## 2. Results

The isolation of microorganisms using mannitol salt agar (MSA) and Columbia blood agar (CBA) from the participants is shown in Table 1. Bacterial growth on MSA and CBA was observed in 117 (58.5%) participants. While there were no significant differences in CoNS and concurrent CoPS and CoNS carriers, *S. aureus* carriers showed moderate significance. Additionally, the presence of other bacterial species and the absence of bacterial growth differed significantly between the two groups (NHP and HP). These findings highlight the importance of considering specific bacterial carriers when comparing different settings and populations.

A total of 200 participants took part in the study, with 117 (58.5%) belonging to the NHP group and 83 (41.5%) belonging to the HP group. The prevalence of CoPS carriers was 6.5% in NHP and 5.5% in HP, with 24 cases (12% of the total). The *p*-value for CoPS carriers indicated a moderate level of statistical significance (*p* = 0.055). CoNS were found in 6% of NHP participants and 24% of HP participants, resulting in 60 cases (30% of the total). The *p*-value for CoNS carriers indicated no statistically significant difference between the groups (*p* = 0.240). Concurrent infections of both CoPS and CoNS were observed in 45.5% of NHP participants and 9% of HP participants, totaling 109 cases (54.5% of the total). The *p*-value for concurrent CoPS and CoNS carriers indicated no statistically significant difference (*p* = 0.090).

Regarding other bacterial species, no NHP participants tested positive, while one HP participant (0.5% of the total) showed growth by other bacteria. The *p*-value for other bacteria indicated a statistically significant difference between the NHP and HP groups (*p* = 0.005). Additionally, one NHP participant (0.5%) did not exhibit bacterial growth, while five HP participants (2.5%) showed no bacterial growth. Six participants (3% of the total) had no bacterial growth. The *p*-value for the prevalence of no bacterial growth indicated a statistically significant difference between the NHP and HP groups (*p* = 0.025).

Table 2 reports the results of the antimicrobial susceptibility test and oxacillin E-test of the isolated *S. aureus*. In the E-test method section, the results showed that 72 (69.2%) of the 104 NHP-CoPS were sensitive to oxacillin, while 20 (19.2%) and 12 (11.5%) were resistant and intermediate, respectively. Similarly, 17 (58.6%) of the 29 HP coagulase-positive staphylococci (HP-CoPS) were sensitive to oxacillin, with 10 (34.5%) and 2 (6.9%) being resistant and intermediate, respectively. The *p*-value associated with this comparison was 0.02, indicating that there was a significant difference in oxacillin susceptibility between the two groups.

The disk diffusion method section shows the susceptibility of the isolated *S. aureus* to different antibiotics. Among the β-lactam antibiotics tested, cefoxitin showed the highest sensitivity. Specifically, 56% of NHP-CoPS and 30.8% of HP-CoPS isolates were sensitive to cefoxitin. These findings indicate that cefoxitin may be more effective in treating NHP-CoPS than HP-CoPS. Furthermore, the *p*-value of 0.03 indicates a significant difference in susceptibility to cefoxitin between the two groups. This implies that there is a notable variation in the response to cefoxitin treatment depending on whether the infection was NHP- or HP-acquired. It is worth noting that resistance to penicillin was observed in all NHP-CoPS and HP-CoPS isolates. This means that neither group showed susceptibility to penicillin, highlighting the importance of using alternative antibiotics for these infections. Among the non-β-lactam antibiotics tested, all NHP-CoPS and HP-CoPS isolates were sensitive to trimethoprim/sulfamethoxazole, and the majority were sensitive to clindamycin. Sensitivity to rifampin and quinupristin was observed in all of the NHP-CoPS and HP-CoPS isolates, while sensitivity to vancomycin was observed in 95.2% and 55.2% of the NHP-CoPS and HP-CoPS isolates, respectively. The *p*-value for this comparison is <0.00001. Similarly, all isolates in both groups were sensitive to trimethoprim/sulfamethoxazole, rifampin, and quinupristin, resulting in *p*-values of <0.00001, 0.00001, and 0.001, respectively.

Table 3 presents the results of detecting genes associated with virulence factors in suspected isolated MRSA strains using PCR techniques. The first genes tested were the *mec* genes associated with methicillin resistance. Both NHP-MRSA and HP-MRSA samples tested positive for the *mecA* and *SCCmecVIb* genes, with 82.9% of all samples testing positive for at least one of the two genes. None of the samples tested positive for the *SCCmecVIa* gene.

The second set of genes analyzed included the *coa* genes, which encode for coagulase, an enzyme produced by MRSA that contributes to its pathogenicity. Additionally, the coa gene was also tested in the samples. All NHP-MRSA and HP-MRSA samples tested positive for at least one size variant of the gene, with the 81-bp and 720-bp variants being the most common. Overall, 100% of the NHP-MRSA samples and 88.2% of the total samples tested positive for at least one variant of the *coa* gene.

The third set of genes analyzed included the *spa* genes, which encode *Staphylococcus* protein A (SpA), a gene associated with the virulence factor in MRSA. While no NHP-MRSA samples tested positive for any gene variants, 64% of the HP-MRSA samples tested positive for the 1100 bp variant. Overall, 8.8% of the total samples tested positive for at least one variant of the *spa* gene.

The last genes analyzed were *aae*, *aap*, *emb*, *IcaD*, and *Luk-PV*, previously associated with MRSA virulence. The *aae* and *aap* genes associated with adhesion and invasion were in NHP and HP-MRSA samples. The *emb* gene, associated with biofilm production, was present in all samples, with the 50-bp variant being the most common. The *IcaD* gene, associated with the biofilm formation, was found in both NHP-MRSA and HP-MRSA samples, with the 200 bp variant being the most common. Finally, the *Luk-PV* gene, associated with virulence, was not detected in any of the samples.

## 3. Discussion

*Staphylococcus aureus*, a coagulase-positive staphylococcal species, is a well-known human pathogen that causes several infections. Over the years, the emergence and spread of antibiotic resistance among *S. aureus* strains have posed significant challenges for healthcare professionals. Furthermore, the presence of virulent factors enhances the pathogenic potential of these bacteria.

Several previous studies, employing various diagnostic methods and microbiological media, have reported similar results to those presented in Table 1, when isolating microorganisms from both NHP and HP carriers [30,31,32]. These studies have consistently found a higher prevalence of CoNS in HP than NHP. For instance, Monegro et al. [30] found a significantly higher proportion of CoNS in HP, corroborating the results in Table 2. Similarly, Downing et al. [31] reported a higher prevalence of CoNS in HP than in NHP. Moreover, previous studies have also consistently reported a higher proportion of *S. aureus* in both NHP and HP, aligning with the results presented in Table 1. For instance, Cheung et al. [32] identified *S. aureus* as the most common cause of both NHP and HP.

When considering the antimicrobial susceptibility testing, the data indicated that NHP-CoPS exhibited higher sensitivity to oxacillin than HP-CoPS. This finding is consistent with previous studies [26,32,33,34,35] indicating that MRSA strains isolated from hospitals are more likely to be oxacillin resistant. In hospitals, there is often a higher prevalence of healthcare-associated infections that may be caused by more resistant strains of *S. aureus*. Prolonged and frequent exposure to antibiotics in the hospital environment can contribute to the selection and dissemination of drug-resistant strains, including MRSA [36]. Additionally, hospitals typically have a higher intensity of antimicrobial usage due to the presence of critically ill patients, invasive medical procedures, and prolonged hospital stays, which can contribute to the emergence and spread of resistant strains. On the other hand, in the community, the usage of antibiotics is generally lower and more restricted to the treatment of acute infections. The lower overall selective pressure from antibiotic use in the community may result in a lower prevalence of resistant strains. While specific data on oxacillin consumption in hospitals and the community were not directly assessed in this study, it is plausible that the higher resistance to oxacillin in hospitals is associated with higher antibiotic consumption in those settings [32,37]. However, further studies specifically investigating the antibiotic consumption patterns and their impact on resistance development are needed to establish a direct relationship.

Several previous studies have reported similar findings when assessing the antimicrobial susceptibility of *S. aureus* [38,39,40]. For instance, a study by Cella et al. [39] found that 71.4% of *S. aureus* isolates were sensitive to oxacillin, while 28.6% were resistant. These results are comparable to the findings of the current study where 69.2% of isolates were sensitive and 30.8% were resistant to oxacillin. Another study by Qodrati et al. [40] revealed that 70.9% of *S. aureus* isolates were sensitive to cefoxitin, with a resistance rate of 29.1%. This is consistent with the results reported in the current study, where 56.7% of isolates were sensitive and 43.3% were resistant to cefoxitin. In addition, Shahid et al. [41] found that 98.6% of *S. aureus* isolates were sensitive to vancomycin, which aligns with the results of the current study, where all isolates were sensitive to vancomycin.

On the other hand, previous studies reported different findings when considering the antimicrobial susceptibility of *S. aureus* in comparison with our study. For example, Salas et al. [42] reported a high prevalence of MRSA among clinical isolates in a Spanish hospital, with resistance rates of 68.4% for cefoxitin and 57.9% for clindamycin. Furthermore, Ferreira et al. [43] reported a high prevalence of *PVL*-positive methicillin-sensitive S. aureus (MSSA) strains in Europe, which showed higher susceptibility to non-β-lactam antibiotics such as trimethoprim/sulfamethoxazole and fusidic acid compared to *PVL*-negative strains. Additionally, Isozumi et al. [44] reported a high prevalence of macrolide-resistant *S. aureus* in Japan, with the *ermB* gene being the most common resistance mechanism. Wang et al. [45] reported a high prevalence of *S. aureus* isolates resistant to multiple antibiotics in China, with the highest resistance rates observed for clindamycin (60.2%), erythromycin (60.2%), and ciprofloxacin (59.1%).

The results of the PCR showed that all NHP-acquired MRSA and HP-acquired MRSA samples tested positive for at least one variant of the *coa* gene, which encodes for coagulase, an enzyme produced by MRSA strains that are typically considered opportunistic pathogens, meaning they can cause infections in individuals who are predisposed or immunocompromised. However, there are instances where MRSA colonization can occur without causing overt infection or symptoms. In these cases, the MRSA strains present may not exhibit the same level of virulence or pathogenicity as those causing clinically clear infections [46]. Additionally, both NHP-acquired MRSA and HP-acquired MRSA samples showed the presence of the *mecA* and *SCCmecVIb* genes, which are associated with methicillin resistance [39,40,41,47,48]. The presence of the *mecA* gene alone does not conclusively indicate a phenotypic resistance profile. Therefore, the detection of these genes alone does not necessarily confirm the resistance of the MRSA strains tested to methicillin and other related antibiotics. This finding is consistent with the observed resistance patterns in antimicrobial susceptibility testing [11]. Several studies have reported similar findings regarding gene detection in NHP- and HP-acquired MRSA. For instance, Larsen et al. [47] found that NHP-acquired MRSA strains were more susceptible to non-β-lactam antibiotics compared with HP-acquired MRSA strains, and both types of MRSA had similar *mecA* and *SCCmec* gene profiles. Similarly, Snitser et al. [11] reported that NHP-acquired MRSA strains were more susceptible to non-β-lactam antibiotics, and both types of MRSA had similar *mecA* gene profiles. Lakhundi et al. [48] also found that NHP-acquired MRSA strains were more susceptible to non-β-lactam antibiotics, and both types of MRSA had similar *SCCmec* gene profiles. However, the literature also includes previous work that reports different findings. For example, Bai et al. [49] found that NHP-acquired MRSA strains were more susceptible to multiple antibiotics compared with HP-acquired MRSA strains, and NHP-acquired MRSA strains had a lower prevalence of PVL gene than HP-acquired MRSA strains. Additionally, Tsouklidis et al. [39] found that HP-acquired MRSA strains were more resistant to multiple antibiotics than NHP-acquired MRSA strains, and HP-acquired MRSA strains had a higher *PVL* gene prevalence than NHP-acquired MRSA strains.

The limitations of this study include the limited generalizability of the results. This study was conducted in a specific geographical region, and the results may not be representative of other regions or countries. As such, variations in bacterial populations and antimicrobial resistance patterns in different geographic locations should be considered. Furthermore, the study participants were selected from specific healthcare settings, which may introduce selection bias. As a result, the findings may not fully represent the entire population of individuals with NHP and HP.

Future studies can explore a larger and more diverse population to provide a broader understanding of the prevalence and resistance patterns of *S. aureus* infections. Additionally, based on the findings of this study, future research can evaluate the effectiveness of targeted intervention strategies to control and prevent *S. aureus* infections. This may include developing and implementing antimicrobial stewardship programs, infection control measures, and surveillance systems to mitigate the impact of multidrug-resistant strains.

## 4. Materials and Methods

### 4.1. Sample Collection

The samples in this study were collected over a period of one year, from September 2018 to August 2019. Two hundred nasal swabs were collected from two different carriers in Riyadh, Saudi Arabia. At King Abdulaziz Medical City (KAMC), Riyadh, Saudi Arabia, 83 nasal swabs were collected from various hospital personnel, including physicians, nurses, radiologists, laboratory medical specialists, echocardiograph technologists, medical residents, hospital administrators, clinical nutritionists, internship students, and medical research assistants. At King Saud bin Abdulaziz University for Health Sciences (KSAU-HS) in Riyadh, Saudi Arabia, 117 nasal swabs were collected from non-hospital personnel, including academic faculties, administrators, and students. The study excluded part-time personnel at KAMC, students with direct patient contact, personnel working part-time at KAMC, and individuals with incomplete information records from the NHP and HP. Each participant was asked to fill out an approved consent form and the questionnaire prepared for this study.

### 4.2. Specimen Collection and Transport

Dry polyester swabs were inserted into each nostril and remained for a few seconds to collect the specimen. The collected nasal swabs were immediately transported to the laboratory. If transportation was not possible within 36 h, they were stored at 2–8 °C for a maximum of five days. Nasal swabs were collected daily during working days for up to three months.

### 4.3. Isolation and Identification of Coagulase-Positive and Negative Staphylococcus

To identify CoPS and CoNS, the collected nasal swabs were screened and identified according to the standard laboratory protocols. The specimens were directly streaked onto selective media that promote the growth of staphylococci, such as mannitol salt agar (MSA) for CoPS and Columbia blood agar (CBA) for CoNS. The isolates obtained from the selective media were subjected to further identification tests. These tests included biochemical tests such as catalase, coagulase, and other specific enzyme tests that help in differentiating CoPS from CoNS and other bacteria.

The CoPS isolates were tested for methicillin resistance using a standard oxacillin salt agar plate procedure and cefoxitin susceptibility, as indicated by the Clinical and Laboratory Standards Institute (CLSI) [50]. Additionally, chromogenic agar was used for rapid MRSA screening, considering its high sensitivity for MRSA. Confirmation of methicillin resistance was performed using a bacterial suspension inoculated onto Mueller–Hinton agar supplemented with calcium (50 mg/L), magnesium (25 mg/L), sodium chloride (4%), and oxacillin (4 g/mL). The growth of at least one well-defined colony on these plates incubated at 30 °C for 48 h indicated MRSA. The methicillin-resistant strain was further confirmed using microdilution in Mueller–Hinton broth supplemented with cations and sodium chloride, with a minimal inhibitory concentration (MIC) of oxacillin more than 8 g/mL. An MIC of oxacillin <1 g/mL was considered MSSA, and an MIC of oxacillin between >1 g/mL and <8 g/mL was defined as a methicillin-intermediate staphylococcus (MISA) [51].

### 4.4. Antimicrobial Susceptibility Testing

The antimicrobial susceptibility of isolated CoPS to seven antimicrobials was determined by the agar diffusion method in Mueller–Hinton agar, following the CLSI guidelines [51]. The results were evaluated after incubating at 35 °C for 24 h. The antimicrobial disks used were penicillin (P, 1 μg), cefoxitin (FOX, 30 μg), vancomycin (VA, 30 μg), clindamycin (CC, 2 μg), sulfamethoxazole/trimethoprim (SXT, 25 μg), rifampin (RA, 25 μg), and quinupristin (SYN, 15 μg). The selection of antimicrobials in this study was based on their relevance to the treatment of *S. aureus* infections and their commonly encountered usage in clinical settings. We focused on antimicrobials such as oxacillin, cefoxitin, and vancomycin because they are commonly used to treat *S. aureus* infections and represent different classes of antibiotics. Additionally, these antimicrobials have established breakpoints and interpretive criteria for susceptibility testing, allowing for an accurate determination of resistance patterns.

Fresh subcultures of isolates were prepared on mannitol salt agar, and the inoculum was adjusted to 0.5 McFarland. A sterile cotton swab was then used to spread the inoculum over the surface of a Mueller–Hinton agar plate, and the antimicrobial disks were applied.

### 4.5. Polymerase Chain Reaction (PCR) Analysis

PCR was performed to detect the presence of toxin genes (*PVL*, *spa*, *coa*, *aae*, *aap*, *emb*, and *IcaD*) and antibiotic-resistant genes (*mecA*, *SCCmecII*, and *SCCmecIVa/b*). DNA was extracted from the isolates using a commercial kit (Qiagen, Valencia, CA, USA) following the manufacturer’s instructions. PCR was conducted using specific primers and previously described PCR conditions (Table 4).

### 4.6. Statistical Analysis

Statistical analysis was conducted using SPSS (v.22.0) software. The differences between hospital-acquired and community-acquired cases were assessed using several variables. Chi-square tests were used to determine the statistical significance of the observed differences, with a *p*-value threshold of <0.05.

### 4.7. Ethical Considerations

The Institutional Review Board at King Abdullah International Medical Research Centre, Riyadh, Saudi Arabia, granted permission to conduct the project after reviewing the ethical aspects of the proposal (Study number RYD-16-417780-75574) (Protocol Code RC13/240 and 19 February 2015).

## 5. Conclusions

This study highlights the high prevalence of coagulase-positive staphylococci (CoPS) and coagulase-negative staphylococci (CoNS) in both non-hospital and hospital personnel carriers. The differences in the distribution of these strains suggest potential occupational risk factors. The findings also reveal varying antibiotic susceptibility patterns, with oxacillin and cefoxitin demonstrating effectiveness against CoPS strains. However, the presence of methicillin resistance genes in both non-hospital and hospital personnel MRSA strains underscores the urgent need for effective management strategies. Further research is warranted to explore additional virulence-associated genes and develop comprehensive approaches for preventing, diagnosing, and treating CoPS infections in Saudi Arabia.

## Figures and Tables

**Table 1 antibiotics-12-01147-t001:** Prevalence and significance of bacterial infections in non-hospital personnel vs. hospital personnel settings.

	Non-Hospital Personnel (NHP) *n* (%)	Hospital Personnel (HP) *n* (%)	Total *n* (%)	*p*-Value
Number of participants	117 (58.5)	83 (41.5)	200 (100)	0.007
Bacterial isolation				
*Staphylococcus aureus* (CoPS)	13 (6.5)	11 (5.5)	24 (12)	0.055
Coagulase-negative staphylococci (CoNS)	12 (6)	48 (24)	60 (30)	0.240
CoPS and CoNS	91 (45.5)	18 (9)	109 (54.5)	0.090
Other bacteria	0	1 (0.5)	1 (0.5)	0.005
No bacterial growth	1 (0.5)	5 (2.5)	6 (3)	0.025

NHP: non-hospital personnel; HP: hospital personnel; CoPS: coagulase-positive staphylococci; CoNS: coagulase-negative staphylococci.

**Table 2 antibiotics-12-01147-t002:** Antimicrobial susceptibility test and oxacillin E-test of isolated *Staphylococcus aureus*.

Variables	NHP-CoPS (*n* = 104) [*n* (%)]	HP-CoPS (*n* = 29) [*n* (%)]	*p*-Value
E-Test Method (MIC):			
Oxacillin			0.20
Resistance	20 (19.2%)	10 (34.5%)	
Intermediate	12 (11.5%)	2 (6.9%)	
Sensitive	72 (69.2%)	17 (58.6%)	
β-Lactam Antibiotics:			
FOX30			0.03
Resistance	30 (28.9%)	11 (38.5%)	
Intermediate	15 (14.4%)	9 (30.8%)	
Sensitive	59 (56.7%)	9 (30.8%)	
P10			<0.00001
Resistance	104 (100%)	29 (100%)	
Intermediate	0	0	
Sensitive	0	0	
Non-β-Lactam Antibiotics:			
VA30			0.01
Resistance	0	0	
Intermediate	5 (4.8%)	13 (46.2%)	
Sensitive	99 (95.2%)	16 (55.2%)	
CC2			<0.00001
Resistance	10 (9.6%)	0	
Intermediate	10 (9.6%)	0	
Sensitive	84 (81%)	29 (100%)	
SXT25			<0.00001
Resistance	0	0	
Intermediate	0	0	
Sensitive	104 (100%)	29 (100%)	
RA25			0.001
Resistance	5 (4.8%)	0	
Intermediate	0	0	
Sensitive	99 (95.2%)	29 (100%)	
SYN15			<0.00001
Resistance	0	0	
Intermediate	0	0	
Sensitive	104 (100%)	29 (100%)	

Penicillin 10 µg (P10); cefoxitin 30 µg (FOX30); vancomycin 30 µg (VA30); clindamycin 2 µg (CC2); sulfmeth/trimeth 25 µg (SXT25); rifampin 25 µg (RA25); quinupristin 15 µg (SYN15). NHP-CoPS: non-hospital personnel-acquired coagulase-positive staphylococci; HP-CoPS: hospital personnel-acquired coagulase-positive staphylococci.

**Table 3 antibiotics-12-01147-t003:** Detected genes in suspected isolated MRSA using PCR.

Gene	NHP-MRSA *n* = 30	HP-MRSA *n* = 11	Total *n* = 41
1. Antibiotic resistance genes			
*mec* genes			
*mecA* (300 bp)	25 (83.3%)	9 (81.8%)	34 (82.9%)
*SCCmecII* (495 bp)	0	0	0
*SCCmecVIa* (21–67 bp)	0	0	0
*SCCmecVIb* (21–67 bp)	25 (83.3%)	9 (81.8%)	34 (82.9%)
	NHP-MRSA *n* = 25	HP-MRSA *n* = 9	Total *n* = 34
2. Genes associated with virulence factors			
*coa* gene			
*coa* (81 bp)	25 (100%)	9 (100%)	34 (100%)
*coa* (120 bp)	1 (4%)	4 (44.4%)	5 (14.7%)
*coa* (400 bp)	0	2 (22.2%)	2 (5.9%)
*coa* (720 bp)	2 (8%)	2 (22.2%)	4 (11.8%)
*Spa* gene			
*spa* (800 bp)	0	3 (33.3%)	3 (8.8%)
*spa* (1020 bp)	0	1 (11.1%)	1 (2.9%)
*spa* (1100 bp)	16 (64%)	3 (33.3%)	19 (55.9%)
*spa* (1120 bp)	0	1 (11.1%)	1 (2.9%)
Other genes			
*aae* (110 bp)	0	2 (22%)	2 (5.9%)
*aae* (220 bp)	6 (24%)	0	6 (17.7%)
*aap* (180 bp)	6 (24%)	0	6 (17.7%)
*aap* (200 bp)	6 (24%)	0	6 (17.7%)
*aap* (300 bp)	6 (24%)	0	6 (17.7%)
*aap* (460 bp)	6 (24%)	0	6 (17.7%)
*aap* (480 bp)	6 (24%)	0	6 (17.7%)
*emb* (50 bp)	25 (100%)	9 (100%)	34 (100%)
*emb* (480 bp)	6 (24%)	0	6 (17.7%)
*IcaD* (100 bp)	0	1 (11.1%)	1 (2.9%)
*IcaD* (200 bp)	13 (44.8%)	3 (33.3%)	16 (47.1%)
*Luk-PV* (433 bp)			

**Table 4 antibiotics-12-01147-t004:** Primers used in PCR.

Gene	Primers	Size	PCR Conditions	References
*mecA*	F 5′-TGGCTATCGTGTCACAATCG-3′ R 5′-CTGGAACTTGTTGAGCAGAG-3′	300 bp	95 °C → 5 min 95 °C → 30 s 56 °C → 40 s 72 °C → 45 s 72 °C → 10 min 4 °C → _∞_	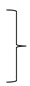	35×	[52]
*SCCmecII*	F 5′-CAAAAGGACTGGACTGGAGTCCAAA-3′ R 5′-CAAGTGAATTGAAACCGCCT-3′	287 bp	[53]
*SCCmecVIa*	F 5′-TTTGAATGCCCTCCATGAATAAAAT-3′ R 5′-AGAAAAGATAGAAGTTCGAAAGA-3′	776 bp
*SCCmecVIb*	F 5′-AGTACATTTTATCTTTGCGTA-3′ R 5′-AGTCATCTTCAATATCGAGAAAGTA-3′	1000 bp
*coa*	F 5′-CGAGACCAAGATTCAACAAG-3′ R 5′-AAAGAAAACCACTCACATCA-3′	81–720 bp	95 °C → 2 min 95 °C → 30 s 58 °C → 2 min 72 °C → 2 min 72 °C → 10 min 4 °C → _∞_	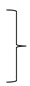	30×	[54]
*spa*	F 5′-ATCTGGTGGCGTAACACCTG-3′ R 5′-CGCTGCACCTAACGCTAATG-3′	800–1120 bp	95 °C → 5 min 95 °C → 30 s 56 °C → 40 s 72 °C → 45 s 72 °C → 10 min 4 °C → _∞_	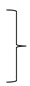	35×	[55]
*aae*	F 5′-AACAAATTGATAAAGCAACG-3′ R 5′-GTTGTCTTTCCTTTAGTGTC-3′	110 and 220 bp	95 °C → 10 min 95 °C → 10 s 55 °C → 20 s 72 °C → 25 s 72 °C → 10 min 4 °C → _∞_	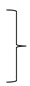	45×	[56]
*aap*	F 5′-TCACTAAACAACCTGTTGACG AA-3′ R 5′-AATTGATTTTTATTATCTGTTGAA TGC-3′	180–480 bp	[57]
*emb*	F 5′-AGCGGTACAAATGTCAATATC-3′ R 5′-AGAAGTGCTCTAGCATCATCC-3′	50–480 bp	96 °C → 2 min 94 °C → 1 min 55 °C → 30 s 72 °C → 1 min 72 °C → 10 min 4 °C → _∞_	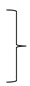	40×	[58]
*IcaD*	F 5′-ATGGTCAAGCCCAGACAGAG-3′ R 5′-CGTGTTTTCAACATTTAATGCAA-3′	100–200 bp	94 °C → 5 min 94 °C → 30 s 55 °C → 30 s 72 °C → 30 s 72 °C → 1 min 4 °C → _∞_	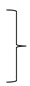	50×	[59]
*Luk-PV*	F 5′-ATCATTAGGTAAAATGTCTGGACATGATCCA-3′ R 5′-GCATCAACTGTATTGGATAGCAAAAGC-3′	433 bp	95 °C → 5 min 95 °C → 30 s 55 °C → 40 s 72 °C → 45 s 72 °C → 10 min 4 °C → _∞_	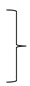	35×	[60]

## Data Availability

The datasets used and/or analyzed during the current study are available from the corresponding author on reasonable request.

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
