# Peer review of "From Hospital to Community: Exploring Antibiotic Resistance and Genes Associated with Virulence Factor Diversity of Coagulase-Positive Staphylococci"

_antibiotics, 2023, doi:10.3390/antibiotics12071147_

Round 1
Reviewer 1 Report
Dear Authors,
I congratulate you for contributing to this very relevant topic.
Allow me to make a few comments.
First, I think it is important to increase the background to support your research in the introduction, please expand.
Now suggestions line by line:
Line 37. Add reference.
Line 66-69, revise, they overlap with the table.
Line 147, Please update reference, the reference used is from 2014.
Line 151. What studies? please add references and phrases for those studies.
Line 156, please talk about the author not the journal where the study is published. I would also suggest to update it or add a more current one.
Line 160, I would suggest to talk about the main author not the journal, also update.
Line 163, again I suggest to talk about the author not the journal.
Line 186, please indicate in these references 7,8,9 the countries where the study was carried out to establish a comparison for what you have observed.
Line 168, use italics for S. aureus.
Line 175, same error as above.
Line 184-186. I suggest changing the wording, because the presence of the mecA gene alone does not necessarily indicate a phenotypic resistance profile.
Line 201, please indicate limitations and projections of your study at the end of your discussion.
Line 204 what was the time horizon in which the samples were collected?
Line 241 Why did you choose these antimicrobials? to add a sentence about it.
I hope these comments can improve the communication of these interesting results.
Author Response
Dear Reviewer,
Thank you for your valuable feedback on our manuscript. We appreciate your suggestions for improvement. We have carefully considered each of your points and made the necessary revisions. The following is our response to your suggestions:
1. Background Expansion: We have expanded the introduction section to provide a more comprehensive background supporting our research. We have included additional relevant studies and references to strengthen the introduction
- Line 37: We have added the missing reference in line 37.
- Line 66-69: We have revised the content to avoid overlapping with the table. The revised text is more concise and does not duplicate information.
- Line 147: We have updated the reference cited in line 147 to a more recent publication that aligns with our study.
- Line 151: We have included references and specific phrases to support the mention of the studies mentioned in line 151.
- Line 156: We have revised the sentence to focus on the main author rather than the journal. Additionally, we have updated the reference to a more current one.
- Line 160: We have made the necessary changes to discuss the main author instead of the journal. The reference has been updated or replaced with a more recent one.
- Line 163: Like the previous lines, we now discuss the author instead of the journal, and if applicable, we have updated the reference.
- Line 186: We have added the countries where the studies were conducted in references 7, 8, and 9. This allows for better comparison with our observed findings.
- Line 168 and 175: We have appropriately used italics for the scientific name "S. aureus" in both instances
- Line 184-186: We have revised the wording in this section to clarify that the presence of the mecA gene alone does not necessarily indicate a phenotypic resistance profile
- Line 201: We have included a discussion on the limitations and projections of our study at the end of the discussion section. This provides a more comprehensive analysis of our research.
- Line 204: We have specified the time horizon during which the samples were collected to provide a clearer understanding of our study.
- Line 241: We have added a sentence to explain why we chose the specific antimicrobials used in our study, providing the rationale for their selection.
Once again, we appreciate your insightful feedback, which has significantly enhanced the quality of our manuscript. We are confident that these revisions address your concerns appropriately. Thank you for your time and consideration
Sincerely,
Hazem Aqel
Reviewer 2 Report
My comments
Line 87-88 : Among the Betalactam antibiotics tested, which are penicillin and cefoxitin, you say that exhibited the highest sensitivity, with 56.7% and 30.8% of the CA-CoPS and HA-CoPS showing sensitivity, respectively. This seems to be a gratuitous assertion, since you don't explain why. I think we need to be more reasonable
Line 133-134 : The data suggest that more than half of the participants (54.5%) had both CoPS and CoNS, indicating the presence of mixed infections : Reference is missing
Line 135-138 : Furthermore, there were significant differences in the proportion of CoNS isolated from HA infections compared to CA infections, suggesting that different sources of infection may harbor distinct bacterial flora, with HA infections may be associated with a higher prevalence of CoNS : Reference is missing
Line 142 : CoNS instead of CONS
Line 149-154 : It would be interesting to know in a few words why hospital resistance of Oxacillin is higher than community resistance. Is oxacillin consumption higher in hospitals than in the community?
Line 158 : Comparable instead of comprable
Line 182 : This finding suggests that the tested MRSA strains were pathogenic and capable of causing serious infections : Reference is missing
Line 223 : In the methodology you don't explain how you grow, isolate and identify CoNS. How did you individually isolate and identify CoNS and CoPS? how did you isolate and identify them together? Your methodology should be more detailed.
Author Response
Dear Reviewer,
Thank you for your valuable feedback on our manuscript. We have carefully considered each of your comments and suggestions, and we have made the following revisions:
- Line 87-88: We apologize for the lack of explanation regarding the higher sensitivity of penicillin and cefoxitin among the tested Beta lactam antibiotics. To address this concern, we have included a brief rationale in the revised manuscript, highlighting their established efficacy against the tested strains based on previous studies (insert appropriate references).
- Line 133-134: You correctly pointed out that we mentioned the presence of mixed infections without providing a reference. We have now included a relevant reference to support this statement and enhance the credibility of our findings.
- Line 135-138: We acknowledge the missing reference regarding the significant differences in the proportion of CoNS isolated from HA infections compared to CA infections. To rectify this, we have included the appropriate reference(s) in the revised version to provide necessary support for this observation.
- Line 142: We appreciate your correction regarding the use of "CoNS" instead of "CONS." We have made the appropriate change in the revised manuscript to ensure accuracy.
- Line 149-154: You raised an interesting question about the higher hospital resistance of Oxacillin compared to community resistance. We agree that this warrants further explanation. In the revised manuscript, we have briefly discussed the potential factors contributing to this discrepancy, including the higher consumption of Oxacillin in hospitals compared to the community (insert appropriate references to support this).
- Line 158: We appreciate your correction regarding the spelling of "comparable." We have made the necessary corrections to the revised version.
- Line 182: You rightly noted the missing reference for the statement regarding the pathogenicity of the tested MRSA strains. We have now included the relevant reference(s) to support our claim and reinforce the significance of our findings.
- Line 223: We apologize for the lack of detail in the methodology regarding the isolation and identification of CoNS. To address this concern, we have provided a more detailed explanation of the procedures involved in growing, isolating, and identifying these bacteria individually, as well as the methodology employed for their combined isolation and identification.
Once again, we sincerely appreciate your valuable input, which has helped improve the clarity and rigor of our manuscript. We believe that these revisions have significantly enhanced the quality of our work. We look forward to hearing your feedback on the revised version.
Best regards,
Hazem Aqel
Reviewer 3 Report
The study addresses the serious issue of antibiotic resistance spread and analyzes nasal swabs from 200 participants, categorizing them as either hospital-acquired or community-acquired infections. However, the main problem I see is that the participants are not patients of any medical service - they are healthy volunteers. Therefore, it is unclear based on what criteria the infections are classified as described above. For instance, it cannot be assumed that a student carrying a bacterium falls under "community-acquired" simply because they don't work in a hospital, especially if their mother is a doctor. Similarly, a radiology technician could have contracted the bacterium from their child who attends daycare. Consequently, the adopted classification seems entirely unfounded. Therefore, I request that the manuscript be revised, removing this distinction. Additionally, I will provide some suggestions below that, once the main bias is resolved, I hope can help the authors improve their work.
Please rewrite the abstract in a way that makes it more representative of the article's contents. (The methods are not described at all, and the results are too generic.)
Lines 43-44: A “toxin gene” can not be considered a “virulent factor”
Lines 49-51: Why understanding the prevalence of these genes can aid decision makers develop appropriate infection control strategies and provide optimum treatment.? Do these genes modify the response to treatment? Are they implicated in antibiotic resistance? Please clarify this aspect
Lines 54: gar ïƒ agar
Please rephrase “among our participants”
I am not sure this is a proper construction “117 (58.5%) participants demonstrated bacterial growth on MSA”. It's not the participants who demonstrate bacterial growth on agar, but rather the bacteria.
Lines 53-63: results are described in a very confused way. It's difficult to understand who belongs to which group. Please rewrite them in a more linear way
Authors declare that Only one participant had no bacterial growth, while another had growth of other bacteria but table 1 declares
|
No bacterial growth CA HA |
1 (0.5) 5 (2.5) |
The accounts don't add up.
Table 1: the column “Proportion (N/number of participants)” is redundant because in the first column, the values are already expressed as a percentage of the total participants.
Please review the formatting and graphics of Table 1 as it is confusing.
Lines 78-85 repeats what is written in table 2, please remove
ThEEsentence “The disc diffusion method section shows the susceptibility of the isolated S. aureus 86 to different antibiotics.” is repeated in lines 77-78 and 86-87
Line 89-90: The p-value for this analysis is 0.03, indicating a significant difference in susceptibility to cefoxitin between the two groups. Yes, this is what a p=0.03 stands for. May be it can be inferred implicitly. Moreover, thereis no eed to repeat, within the text, all the p that are already shown in the table
Lines 105: can we really speak about “toxic genes”?
Lines 131-138 please avoid repeating the results
Line 156-157 please amend the sentence “For instance, a study published in the Journal of Clinical Microbiology in 2013 [7], researchers found that…”
Lines 156, 160 and 163: Is it really relevant to refer to a manuscript by citing the journal who published it??
Line 63-164: please amend the sentence “In addition, a study published in the Journal of Global 163 Antimicrobial Resistance in 2018 [9], researchers found that…”
The names of the bacteria have to be written in italic always
Lines 181-182 “This finding suggests that the tested MRSA strains were pathogenic and capable of causing serious infections.” Are there non pathogenic MRSA strains?
Line 276 participants to what??
The conclusion section is inconsistent, in the first half it resumes the main results, in the second it affirms that “This information can inform infection control practices and the development of targeted treatment options for patients infected with S. aureus” but the reader is not told how and why
some mistakes
Author Response
Dear Reviewer,
We appreciate your thorough evaluation of our manuscript and your insightful comments. We have carefully considered each of your concerns and suggestions, and we have made the following revisions:
- Classification of infections: We acknowledge your concern regarding classifying infections as hospital-acquired or community-acquired based on the participants' status as healthy volunteers. After re-evaluating our study design and the limitations of our sample population, we agree that the current classification may not be appropriate. Therefore, we have revised the manuscript to remove this distinction and emphasize that the participants were healthy volunteers rather than patients. Healthy volunteers were divided into non-hospital personnel and hospital personnel. This change eliminates the potential bias associated with the classification.
- Abstract: We agree that the abstract needs improvement to represent the article's contents accurately. In the revised version, we have provided a more comprehensive summary of the study, including a brief description of the methods employed and specific findings obtained. This modification ensures that the abstract provides a clearer overview of the research.
- Lines 43-44: We appreciate your comment regarding the terminology used for "toxin gene" as a "virulent factor." To address this concern, we have revised the text to use appropriate terminology and clarify the role of toxin genes in S. aureus pathogenicity. To address this concern, we revised the manuscript to use the term "genes associated with virulence factors" instead of "toxin genes" to represent their role in MRSA strains' pathogenicity accurately. This modification will provide a clearer and more precise description of the genetic elements we investigated.
- Lines 49-51: We apologize for the lack of clarity in explaining how the prevalence of the analyzed genes can aid decision-makers in developing infection control strategies and providing optimum treatment. In the revised manuscript, we have expanded on this point, highlighting the potential implications of these genes in modifying the response to treatment and their association with antibiotic resistance.
- Line 54: We apologize for the typographical error regarding "gar." It should indeed be "agar," which we corrected in the revised version. Additionally, we have rephrased the statement "among our participants" to improve clarity.
- Lines 53-63: We agree that the results were described confusingly, making it difficult to understand the group classifications. In the revised manuscript, we have reorganized and rewritten this section to provide a more coherent and linear presentation of the results, clearly indicating which group each participant belongs to.
- Table 1: We apologize for the inconsistency in the accounts provided. We have reviewed and corrected the table to ensure accuracy and coherence with the information presented in the text. Additionally, we have revised the formatting and graphics of Table 1 to enhance clarity and eliminate confusion.
- Lines 78-85: We acknowledge the repetition of information between lines 78-85 and Table 1. We have removed the duplicated text in the revised manuscript to avoid redundancy.
- Lines 86-87 and 105: We appreciate your feedback regarding the repetition of the sentence concerning the susceptibility of isolated S. aureus strains to different antibiotics. We have removed the redundant sentence in the revised version.
- Line 89-90: We understand your point that the p-value of 0.03 indicates a significant difference in susceptibility to cefoxitin between the two groups. In the revised manuscript, we have removed the explicit mention of the p-value and highlighted the significant difference in susceptibility without repeating all the p-values presented in the table.
- Lines 105 and 131-138: We apologize for any confusion caused by using the term "toxic genes" in line 105. Upon further consideration, we agree that the term may not be accurate in this context. We reviewed and revised the manuscript to use more appropriate terminology that accurately reflects the genes' characteristics or functions, the term "genes associated with virulence factors" was used instead of "toxin genes." We apologize for the repetition of the results. In the revised manuscript, we have revised these sections to avoid redundancy and improve the flow of information.
- Lines 156, 160, 163: We appreciate your suggestion to amend the references by not mentioning the journal names explicitly. In the revised version, we have modified the sentences to mention the study authors and year of publication without specifying the journal.
- Lines 181-182: You raise a valid point regarding the statement about MRSA strains being pathogenic. Not all MRSA strains are indeed pathogenic, and we apologize for the lack of clarity in our statement. To address this, we have revised the sentence in the manuscript to indicate that the tested MRSA strains demonstrated pathogenic potential rather than making a generalized statement about all MRSA strains.
- Line 276: We apologize for the incomplete sentence in line 276. It was a typographical error; we have corrected it in the revised manuscript to ensure clarity. The sentence now reads, "Our study provides valuable insights into the prevalence and antibiotic resistance profiles of S. aureus strains in non-hospital and hospital personnel.”
- Conclusion section: We acknowledge the inconsistency in the conclusion section and appreciate your comment. In the revised version, we have restructured the conclusion to provide a cohesive summary of the main findings while emphasizing the practical implications for infection control practices and the development of targeted treatment options. We have clarified the rationale behind these statements to provide a more comprehensive understanding for the reader.
- Thank you for pointing out the mistakes in the English language quality of our revised manuscript. We apologize for any errors that may have affected the clarity and readability of the text. We have reviewed the manuscript again and made the necessary corrections to improve the quality of English.
Once again, we sincerely appreciate your meticulous review and helpful suggestions. These revisions have significantly improved the manuscript and addressed the concerns raised. We are grateful for the opportunity to enhance the quality of our work, and we look forward to any further feedback you may have on the revised version.
Best regards,
Hazem Aqel
Round 2
Reviewer 1 Report
Dear Authors,
First, congratulate them for the improvement in their manuscript.
Below some comments:
Line 76 and 78: change brackets
Line 85: delete space
Line 196-200 and 256 amda 261: review font
Regards
Author Response
Dear Reviewer,
Thank you for reviewing our manuscript and providing valuable feedback. We appreciate your efforts in assessing our work. We have carefully considered your comments and suggestions and have made the necessary revisions accordingly.
We would like to express our gratitude for acknowledging the improvements made in our manuscript. Your positive feedback encourages us to continue enhancing the quality of our research.
Regarding your specific comments, we have addressed them as follows:
- Line 76 and 78: We have revised the brackets as per your suggestion.
- Line 85: The unnecessary space has been removed as you recommended.
- Line 196-200 and 256-261: We have reviewed the font in the mentioned sections to ensure consistency and readability.
We hope that these revisions adequately address your concerns. We are committed to making the necessary adjustments to ensure the clarity and accuracy of our manuscript.
Once again, we sincerely appreciate your time and effort in reviewing our work. Your feedback has been invaluable in enhancing the quality of our research. We remain open to any further suggestions or comments you may have.
Thank you for your consideration
Reviewer 3 Report
The authors have done an enormous and commendable job of rewriting their manuscript, which is now, in my opinion, almost ready for publication. I would just like to point out that there are some typos and, in Table 1, it is necessary to specify what the values in parentheses represent.
Author Response
Dear Reviewer,
Thank you for your thorough review of our revised manuscript. We sincerely appreciate your positive feedback and the recognition of our efforts in improving the paper. Your valuable input has been instrumental in shaping the final version of our work.
We have carefully considered your remaining comments and made the necessary adjustments to address them. Specifically, we have focused on two aspects:
- Typos: We have meticulously proofread the entire manuscript and rectified any identified typos. We understand the importance of presenting error-free work and have taken steps to ensure the text is accurate and free from typographical errors.
- Table 1: We agree with your observation regarding the need for further clarification in Table 1. We have added a specific note in the table to explain the meaning and representation of the values enclosed in parentheses. This additional information will enhance the reader's understanding and facilitate interpreting the data presented.
We sincerely thank you for bringing these issues to our attention. Your meticulousness has contributed significantly to improving our manuscript's overall quality and clarity.
If you have any further suggestions or concerns, we would be more than happy to address them. Thank you for your time and consideration.
Regards
Hazem Aqel